# A Universal Representation Transformer Layer for Few-Shot Image Classification

**Lu Liu**[1,2*]**, William Hamilton**[1,3†]**, Guodong Long**[2]**, Jing Jiang**[2]**, Hugo Larochelle**[1,4†]
[1] Mila, [2] Australian AI Institute, UTS, [3] McGill University, [4] Google Research, Brain Team
Correspondence to `lu.liu.cs@icloud.com`

## Abstract

Few-shot classification aims to recognize unseen classes when presented with only a small number of samples. We consider the problem of *multi-domain* few-shot image classification, where unseen classes and examples come from diverse data sources. This problem has seen growing interest and has inspired the development of benchmarks such as Meta-Dataset. A key challenge in this multi-domain setting is to effectively integrate the feature representations from the diverse set of training domains. Here, we propose a Universal Representation Transformer (URT) layer, that meta-learns to leverage universal features for few-shot classification by dynamically re-weighting and composing the most appropriate domain-specific representations. In experiments, we show that URT sets a new state-of-the-art result on Meta-Dataset. Specifically, it achieves top-performance on the highest number of data sources compared to competing methods. We analyze variants of URT and present a visualization of the attention score heatmaps that sheds light on how the model performs cross-domain generalization. Our code is available at https://github.com/liulu112601/URT.

## 1 Introduction

Learning tasks from small data remains a challenge for machine learning systems, which show a noticeable gap compared to the ability of humans to understand new concepts from few examples. A promising direction to address this challenge is developing methods that are capable of performing transfer learning across the collective data of many tasks. Since machine learning systems generally improve with the availability of more data, a natural assumption is that few-shot learning systems should benefit from leveraging data *across many different tasks and domains*—even if each individual task has limited training data available.

This research direction is well captured by the problem of *multi-domain few-shot classification*. In this setting, training and test data spans a number of different domains, each represented by a different source dataset. A successful approach in this multi-domain setting must not only address the regular challenge of few-shot classification—i.e., the challenge of having only a handful of examples per class. It must also discover how to leverage (or ignore) what is learned from different domains, achieving generalization and avoiding cross-domain interference.

Recently, Triantafillou et al. (2020) proposed a benchmark for multi-domain few-shot classification, Meta-Dataset, and highlighted some of the challenges that current methods face when training data is heterogeneous. Crucially, they found that methods which trained on all available domains would normally obtain improved performance on some domains at the expense of others. Following on their work, progress has been made, which includes the design of adapted hyper-parameter optimization strategies (Saikia et al., 2020) and more flexible meta-learning algorithms (Requeima et al., 2019). Most notable is SUR (Selecting Universal Representation) (Dvornik et al., 2020), a method that relies on a so-called universal representation, extracting from a collection of pre-trained and domain-specific neural network backbones. SUR prescribes a hand-crafted feature-selection procedure to infer how to weight each backbone for each task at hand, and produces an adapted representation for each task. This was shown to lead to some of the best performances on Meta-Dataset.

---

*This work was done while Lu Liu was a research intern with Mila.
†Canada CIFAR AI Chair

In SUR, the classification procedure for each task is fixed and not learned. Thus, except for the underlying universal representation, there is no transfer learning performed with regards to how classification rules are inferred across tasks and domains. Yet, cross-domain generalization might be beneficial in that area as well, in particular when tasks have only few examples per class.

**Present work.** To explore this question, we propose a Universal Representation Transformer (URT) layer, which can effectively learn to transform a universal representation into task-adapted representations. The URT layer is inspired from Transformer (Vaswani et al., 2017) and uses an attention mechanism to learn to retrieve or blend the appropriate backbones to use for each task. By training this layer across few-shot tasks from many domains, it can support transfer across these tasks.

We show that our URT layer on top of a universal representation's pre-trained backbones sets a new state-of-the-art performance on Meta-Dataset. It succeeds at outperforming SUR on 4 dataset sources without impairing accuracy on the others. This leads to top performance on 7 dataset sources when comparing to a set of competing methods. To interpret the strategy that URT learns to weigh the backbones from different domains, we visualize the attention scores for both seen and unseen domains and find that our model generates meaningful weights for the pre-trained domains. A comprehensive analysis on variants and ablations of the URT layer is provided to show the importance of various components of URT, notably the number of attention heads.

## 2 FEW-SHOT CLASSIFICATION

### 2.1 PROBLEM SETTING

In this section, we will introduce the problem setting for few-shot classification and the formulation of meta-learning for few-shot classification. Few-shot classification aims to classify samples where only few examples are available for each class. We describe a few-shot learning classification task as the pair of examples, comprising of a support set $S$ to define the classification task and the query set $Q$ of samples to be classified.

Meta-learning is a technique that aims to model the problem of few-shot classification as learning to learn from instances of few-shot classification tasks. The most popular way to train a meta-learning model is with episodic training. Here, tasks $T = (Q, S)$ are sampled from a larger dataset by taking subsets of the dataset to build a support set $S$ and a query set $Q$ for the task. A common approach is to sample $N$-way-$K$-shot tasks, each time selecting a random subset of $N$ classes from the original dataset and choosing only $K$ examples for each class to add to the support set $S$.

The meta-learning problem can then be formulated by the following optimization:

$$\min_{\Theta} \mathbb{E}_{(S,Q) \sim p(T)} \left[ \mathcal{L}(S, Q, \Theta) \right], \ \mathcal{L}(S, Q, \Theta) = \frac{1}{|Q|} \sum_{(\boldsymbol{x}, y) \sim Q} -\log p(y | \boldsymbol{x}, S; \Theta) + \lambda \Omega(\Theta), \quad (1)$$

where $p(T)$ is the distribution of tasks, $\Theta$ are the parameters of the model and $p(y | \boldsymbol{x}, S; \Theta)$ is the probability assigned by the model to label $y$ of query example $\boldsymbol{x}$ (given the support set $S$), and $\Omega(\Theta)$ is an optional regularization term on the model parameters with factor $\lambda$.

Conventional few-shot classification targets the setting of $N$-way-$K$-shot, where the number of classes and examples are fixed in each episode. Popular benchmarks following this approach include Omniglot (Lake et al., 2015) or benchmarks made of subsets of ImageNet, such as *mini*ImageNet (Vinyals et al., 2016) and *tiered*ImageNet (Ren et al., 2018). In such benchmarks, the tasks for training cover a set of classes that is disjoint from the classes in the test set of tasks. However, with the training and test sets tasks coming from a single dataset/domain, the distribution of tasks found in either sets is similar and lacks variability, which may be unrealistic in practice.

It is in this context that Triantafillou et al. (2020) proposed Meta-Dataset, as a further step towards large-scale, multi-domain few shot classification. Meta-Dataset includes ten datasets (domains), with eight of them available for training. Additionally, each task sampled in the benchmark varies in the number of classes $N$, with each class also varying in the number of shots $K$. As in all few-shot learning benchmarks, the classes used for training and testing do not overlap.

## 2.2 BACKGROUND AND RELATED WORK

**Meta-Learning** A promising approach for few-shot classification is to use meta-learning to more directly train a model to learn to perform few-shot classification, in an end-to-end way. The two most popular methods are Prototypical Networks (Snell et al., 2017) and Model Agnostic Meta-Learning (MAML) (Finn et al., 2017). Triantafillou et al. (2020) showed that prototypical networks and MAML could be combined by leveraging prototypes for the initialization of the output weights value in the inner loop. Requeima et al. (2019) also proposed Conditional Neural Adaptive Processes (CNAPs) for few-shot classification, which can be seen as extending prototypical networks with a more sophisticated architecture that allows for improved task adaptation. This architecture was later improved further by Bateni et al. (2020) with Simple CNAPS, leading to one of the current best methods on Meta-Dataset. Another line of work which leverages the idea of "transfer by fine-tuning" can be found in Appendix B.

**Universal Representations** In contrast, our work instead builds on that of Dvornik et al. (2020) and their method SUR (Selecting from Universal Representations). Bilen & Vedaldi (2017) introduced the term *universal representation* to refer to a representation that supports good performance in multiple domains. One proposal towards such a representation is to train different neural networks backbones separately on the data of each available domain, then simply to concatenate the representation learned by each. Another is to introduce some parameter sharing between the backbones, by having a single network conditioned on the domain of the provenance of each batch of training data (Rebuffi et al., 2018), e.g. using Feature-wise Linear Modulate (FiLM) (Perez et al., 2018). SUR proposes to leverage a universal representation in few-shot learning tasks with a feature selection procedure that assigns different weights to each of the domain-specific subvectors of the universal representation. The objective is to assign high weights only to the domain-specific representations that are specifically useful for each few-shot task at hand. The weights are inferred by optimizing a loss on the support set that encourages high accuracy of a nearest-centroid classifier. As such, the method does not involve any meta-learning—a choice motivated by the concern that meta-learning may struggle in generalizing to domains that are dissimilar to the training domains. SUR achieved some of the best performances on Meta-Dataset. However, a contribution of our work is to provide evidence that meta-learning can actually be used to replace SUR's hand-designed inference procedure and improve performance further.

**Task Adaptive Representations** Another line of work tries to retrieve task adaptive representations for each task. Task specific representations can be conditioned on a representation of the current task (Oreshkin et al., 2018; Wang et al., 2019), projected to another space (Yoon et al., 2019), or masked based on inter-class commonality and inter-class uniqueness (Li et al., 2019). While the representation extracted from URT is also task adaptive, it is adaptive to a set of pretrained backbones and can be applied to more complicated multi-domain scenarios. Wang & Hebert (2016) proposed to improve a CNN by adding extra layers and train it using unsupervised data while our contribution mainly lies in composing representations instead of an improved CNN. Alet et al. (2018) introduced a modular meta-learning method, which learns a repertoire of modules that serves as nodes to construct a tree structure to solve a new robotic-related task. Comparatively, URT is a one-for-all layer which doesn't need to construct different module structures for each task.

**Transformer Networks** Our meta-learning approach to leverage universal representations is inspired directly from Transformer networks (Vaswani et al., 2017). Our model structure is inspired by the structure of the dot-product self-attention in the Transformer, which we adapted here to multi-domain few-shot learning by designing appropriate parametrizations for queries, keys and values. Self-attention was explored in the single-domain training regime by Ye et al. (2020); Liu et al. (2019b;a; 2020), however for a different purpose, where each representation of individual examples in a task support set is influenced by all other examples. Rather than using self-attention between individual examples in the support set, our model uses self-attention to select between different domain-specific backbones.

## 3 UNIVERSAL REPRESENTATION TRANSFORMER LAYER

In this section, we describe our proposed URT layer, which uses meta-learning episodic training to learn how to combine the domain-specific backbones of a universal representation for any given few-shot learning classification task. URT layer can be built on top of any set of pretrained backbones

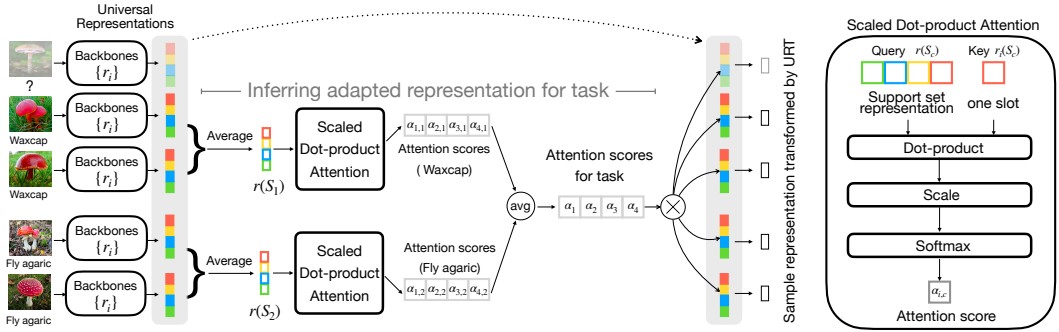

Figure 1: Illustration of how a single-head URT layer uses a universal representation to produce a task-specific representation. This example assumes the use of four backbones, with each color illustrating their domain-specific sub-vector representation in the universal representation.

without further costly fine-tuning of the backbones. More details on how to train multiple domain-specific backbones can be found in Appendix C.

Conceptually, the proposed model views the support set $S$ of a task as providing information on how to query and retrieve from the set $\{r_i\}$ of $m$ pre-trained backbones the most appropriate backbone to build an adapted representation $\phi$ for the task.

We would like the model to support a variety of strategies on how to retrieve backbones. For example, it might be beneficial for the model to retrieve a single backbone from the set, especially if the domain of the given task matches perfectly that of a domain found in the training set. Alternatively, if some of the training domains benefit from much more training data than others, a better strategy might be to attempt some cross-domain generalization towards the few-shot learning task by blending many backbones together, even if none matches the domain of the task perfectly.

This motivates us to use dot-product self-attention, inspired by layers of Transformer networks (Vaswani et al., 2017). For this reason, we refer to our model as a Universal Representation Transformer (URT) layer. Additionally, since each class of the support set might require a different strategy, we perform attention separately for each class and their support set $S_c = \{\boldsymbol{x} | (\boldsymbol{x}, y) \in S \text{ and } y = c\}$.

### 3.1 SINGLE-HEAD URT LAYER

We start by describing an URT layer consisting of a single attention head. An illustration of a single-head URT layer is shown in Figure 1. Let $r_i(\boldsymbol{x})$ be the output vector of the backbone for domain $i$. We then write the universal representation as

$$r(\mathbf{x}) = \text{concat}(r_1(\mathbf{x}), \ldots, r_m(\mathbf{x})). \tag{2}$$

This representation provides a natural starting point to obtain a representation of a support set class. Specifically, we will note

$$r(S_c) = \frac{1}{|S_c|} \sum_{\boldsymbol{x} \in S_c} r(\boldsymbol{x}) \tag{3}$$

as the representation for the set $S_c$. From this, we can describe the URT layer by defining the queries[1], keys, the attention mechanism and output of the layer:

**Queries $\mathbf{q}_c$:** For each class $c$, we obtain a query through $\mathbf{q}_c = \mathbf{W}^q r(S_c) + \mathbf{b}^q$, where we have a learnable query linear transformation represented by matrix $\mathbf{W}^q$ and bias $\mathbf{b}^q$.

**Keys $\mathbf{k}_{i,c}$:** For each domain $i$ and class $c$, we define keys as $\mathbf{k}_{i,c} = \mathbf{W}^k r_i(S_c) + \mathbf{b}^k$, using a learnable linear transformation $\mathbf{W}^k$ and $\mathbf{b}^k$ and where $r_i(S_c) = 1/|S_c| \sum_{\boldsymbol{x} \in S_c} r_i(\boldsymbol{x})$, using a similar notation as for $r(S_c)$.

---

[1]Unable to avoid the unfortunate double usage of the term "query" due to conflicting conventions, we highlight the difference between the *query* sets $Q$ of few-shot tasks and the *queries* $\mathbf{q}_c$ of an attention mechanism.

---
**Algorithm 1** Training of URT layer

---
**Input:** Number of tasks $\tau_{total}$, $m$ pre-trained backbones ;
 1: **for** $\tau \in \{1, \cdots, \tau_{total}\}$ **do**
 2:      Sample a few-shot task $T$ with support set $S$ and query set $Q$;
 3:      **# Infer adapted representation for task from** $S$
 4:      For each class, obtain representation using $m$ pre-trained backbones as in Eq. (3);
 5:      Obtain attention scores using Eq. (4,5) for each head using support set $S$;
 6:      **# Use adapted representation to predict labels in** $Q$ **from support set** $S$
 7:      Compute adapted representation of examples in $S$ and $Q$ as in Eq. (6,7);
 8:      Compute probabilities of label of examples in $Q$ using Prototypical Network as in Eq. (9);
 9:      Compute loss as in Eq. (1,8) and perform gradient descent step on URT parameters $\Theta$;
10: **end for**

---

**Attention scores** $\alpha_i$**:** as for regular Transformer layers, we use scaled dot-product attention

$$\alpha_{i,c} = \frac{\exp(\beta_{i,c})}{\sum_{i'} \exp(\beta_{i',c})}, \beta_{i,c} = \frac{\mathbf{q}_c^\top \mathbf{k}_{i,c}}{\sqrt{l}}, \tag{4}$$

where $l$ is the dimensionality of the keys and queries. Then, these per-class scores are aggregated to obtain scores for the full support set by averaging

$$\alpha_i = \frac{\sum_c \alpha_{i,c}}{N}. \tag{5}$$

Equipped with these attention scores, the URT layer can now produce an adapted representation for the task (for the support and query set examples) by computing

$$\phi(\mathbf{x}) = \sum_i \alpha_i r_i(\mathbf{x}) . \tag{6}$$

As we can see, this approach has the flexibility of either selecting a single domain-specific backbone (by assigning $\alpha_i = 1$ for a single domain) or blending different domains together (by having $\alpha_i >> 0$ for multiple backbones).

## 3.2   MULTI-HEAD URT LAYER

The URT layer described so far can only learn to retrieve a single backbone (or blending of backbones). Yet, it might be beneficial to retrieve multiple different (blended) backbones, especially for a few-shot task that would include many classes of varying complexity.

Thus, to achieve such diversity in the adapted representation, we also consider URT layers with multiple heads, i.e. where each head corresponds to the calculation of Equation 6 and each head has its own set of parameters ($\mathbf{W}^q, \mathbf{b}^q, \mathbf{W}^k, \mathbf{b}^k$). Denoting each head now as $\phi_h$, a multi-head URT layer then produces as its output the concatenation of all of its heads:

$$\phi(\mathbf{x}) = \text{concat}(\phi_1(\mathbf{x}), \ldots, \phi_{\text{H}}(\mathbf{x})). \tag{7}$$

Empirically we found that the randomness in the initialization of head weights alone did not lead to uniqueness and being complimentary between the heads, so inspired by Lin et al. (2017), we add a regularizer to avoid duplication of the attention scores:

$$\Omega(\Theta) = \|(\mathbf{A}\mathbf{A}^\top - \mathbf{I})\|_F^2, \tag{8}$$

where $\| \cdot \|_F$ is the Frobenius norm of a matrix and $\mathbf{A} \in \mathbb{R}^{n \times m}$ is the matrix for attention scores, with $\mathbf{A}_h$ being the vector of all scores $\alpha_i$ for head $h$. The identity matrix $\mathbf{I}$ regularizes each set of attention scores to be more focused so that multiple heads can attend to different domain-specific backbones.

## 3.3   TRAINING STRATEGY

We train representations produced by the URT layer by following the approach of Prototypical Networks (Snell et al., 2017), where the probability of a label $y$ for a query example $\boldsymbol{x}$ given the

Table 1: Test accuracy (mean±CI%95) over 600 few-shot tasks. URT and the most recent methods, which are listed in the first column, are compared on Meta-Dataset (Triantafillou et al., 2020), which are listed in the first row. The numbers in **bold** have intersecting confidence intervals with the most accurate method.

| | ILSVRC | Omniglot | Aircraft | Birds | Textures | QuickDraw | Fungi | VGGFlower | TrafficSigns | MSCOCO | avg. rank |
|---|---|---|---|---|---|---|---|---|---|---|---|
| **MAML** | 37.8±1.0 | 83.9±1.0 | 76.4±0.7 | 62.4±1.1 | 64.1±0.8 | 59.7±1.1 | 33.5±1.1 | 79.9±0.8 | 42.9±1.3 | 29.4±1.1 | 6.4 |
| **ProtoNet** | 44.5±1.1 | 79.6±1.1 | 71.1±0.9 | 67.0±1.0 | 65.2±0.8 | 64.9±0.9 | 40.3±1.1 | 86.9±0.7 | 46.5±1.0 | 39.9±1.1 | 5.7 |
| **ProtoMAML** | 46.5±1.1 | 82.7±1.0 | 75.2±0.8 | 69.9±1.0 | 68.3±0.8 | 66.8±0.9 | 42.0±1.2 | 88.7±0.7 | 52.4±1.1 | 41.7±1.1 | 4.1 |
| **CNAPs** | 50.8±1.1 | 91.7±0.5 | 83.7±0.6 | 73.6±0.9 | 59.5±0.7 | 74.7±0.8 | 50.2±1.1 | 88.9±0.5 | 56.5±1.1 | 39.4±1.1 | 3.6 |
| **SUR** | **56.1±1.1** | 93.1±0.5 | **84.6±0.7** | 70.6±1.0 | **71.0±0.8** | 81.3±0.6 | **64.2±1.0** | 82.8±0.7 | 53.4±1.0 | 50.1±1.0 | 2.3 |
| **SimpleCNAPS** | **56.5±1.1** | 91.9±0.6 | 83.8±0.7 | **76.1±0.8** | 70.0±0.8 | 78.3±0.7 | 49.1±1.2 | **91.3±0.6** | **59.2±1.0** | 42.4±1.1 | 2.2 |
| **URT (Ours)** | **55.7±1.0** | 94.4±0.4 | 85.8±0.6 | 76.3±0.8 | 71.8±0.7 | 82.5±0.6 | 63.5±1.0 | 88.2±0.6 | 51.1±1.1 | **52.2±1.1** | 1.5 |

support set of a task is modeled as:

$$p(y = c|\boldsymbol{x}, S; \Theta) = \frac{\exp(-d(\phi(\boldsymbol{x}) - \boldsymbol{p}_c))}{\sum_{c'=1}^{N} \exp(-d(\phi(\boldsymbol{x}) - \boldsymbol{p}_{c'}))}, \tag{9}$$

where $d$ is a distance metric and $\boldsymbol{p}_c = 1/|S_c| \sum_{\boldsymbol{x} \in S_c} \phi(\boldsymbol{x})$ corresponds to the centroid of class $c$, referred to as its prototype. We use (negative) cosine similarity as the distance. The full training algorithm is presented in Algorithm 1.

## 4 EXPERIMENTS

In this section, we seek to answer three key experimental questions:

**Q1** How does URT compare with previous state-of-the-art on Meta-Dataset for multi-domain few-shot classification?

**Q2** Do the URT attention heads generate interpretable and meaningful attention scores?

**Q3** Does the URT layer provide consistent benefits, even when pre-trained backbones are trained in different ways?

In addition, we investigate architectural choices made, such as our models for keys/queries and their regularization, and study their contribution to achieving strong performance with URT.

### 4.1 DATASETS AND SETUP

We test our methods on the large-scale few-shot learning benchmark Meta-Dataset (Triantafillou et al., 2020). It consists of ten datasets with various data distributions across different domains, including natural images (Birds, Fungi, VGG Flower), hand-written characters (Omniglot, Quick Draw), and human created objects (Traffic Signs, Aircraft). Among the ten datasets, eight provide data that can be used during either training, validation and testing (with each class assigned to only one of those sets), while two datasets are solely used for testing. Following Bateni et al. (2020); Requeima et al. (2019), we also report results on MNIST (LeCun et al., 1998), CIFAR10 and CIFAR100 (Krizhevsky et al., 2009) as additional unseen test datasets. Following Triantafillou et al. (2020), few-shot tasks are sampled with varying number of classes $N$, varying number of shots $K$ and class imbalance. The performance is reported as the average accuracy over 600 sampled tasks. More details of Meta-Dataset can be found in Triantafillou et al. (2020).

The domain-specific backbones are pre-trained following the setup in (Dvornik et al., 2020). Then, we freeze the backbone and train the URT layer for 10,000 episodes, with an initial learning rate of 0.01 and a cosine learning rate scheduler. Following Chen et al. (2020), the training episodes have 50% probability coming from the ImageNet data source. Since different pre-trained backbones may produce representations with different vector norms, we normalize the outputs of the backbones as in Dvornik et al. (2020). URT is trained with parameter weight decay of 1e-5 and with a regularization factor $\lambda = 0.1$. The number of heads ($H$ in Equation 7), is set to 2 and the dimension of the keys and

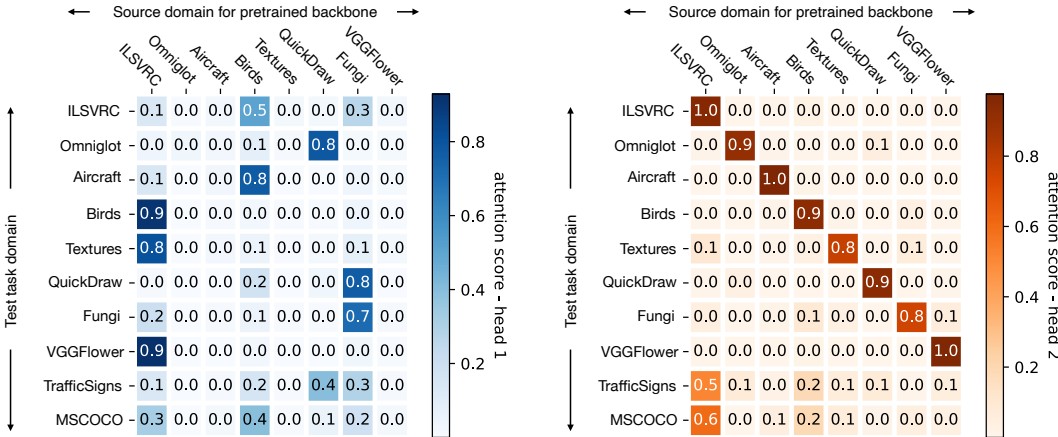

Figure 2: Average attention scores generated by URT with two heads. Rows correspond to the domain of the test tasks and the columns correspond to the pre-trained backbones $r_i(\boldsymbol{x})$ trained on the eight training domains.

queries ($l$ in Equation 4) is set to 1024. We choose the hyper-parameters based on the performance of the validation set. Details of the hyper-parameter selection and how the performance is influenced by them are outlined in Section 4.5. We find that the bottleneck of training URT is extracting features from CNN. Since we freeze the CNN when training the URT, we find dumping the extracted feature episodes can significantly speed up the training procedure from days to around 2 hours.

## 4.2 COMPARISON WITH PREVIOUS APPROACHES

Table 1 presents a comparison of URT with SUR, as well as other baselines based on transfer learning by fine-tuning (Saikia et al., 2020) or meta-learning (Prototypical Networks (Snell et al., 2017), first-order MAML (Finn et al., 2017), ProtoMAML (Triantafillou et al., 2020), CNAPs (Requeima et al., 2019)) and Simple CNAPS(Bateni et al., 2020).

We observe in Table 1 that URT establishes a new state-of-the-art on Meta-Dataset, by achieving the top performance on 8 out of the 10 dataset sources. When comparing to its predecessor, URT outperforms SUR on 4 datasets without compromising performance on others, which is challenging to achieve in the multi-domain setting. Of note, the average inference time for URT is 0.04 second per task, compared to 0.43 for SUR, on a single V100. Thus, getting rid of the optimization procedure for every episode with our meta-trained URT layer also significantly increases the latency, by more than 10×. More results on additional datasets can be found in Appendix A.

## 4.3 INTERPRETING AND VISUALIZING ATTENTION BY URT

To better understand how the URT model of Section 4.2 uses its two heads to build adapted representations, we visualize the attention scores produced on the test tasks of Meta-Dataset in Figure 2.

The blue (first head) and orange (second head) heatmaps summarize the values of the attention scores (Equation 5), averaged across several tasks for each test domain. Specifically, the element on row $t$ and column $i$ is the averaged attention scores $\alpha_i$ computed on test set domain $t$ for the backbone from domain $i$. Note that the last two rows are the two unseen domain datasets. We found that for datasets from the seen domains, i.e. the first eight rows, one head (right, orange) consistently puts most of its weight on the backbone pre-trained on the same domain, while the other head (left, blue) learns relatively smoother weight distributions that blend other related domains. For unseen datasets, the right head puts half of its weight on ImageNet and the left head learned to blend the representations from four backbones.

Table 2: Test accuracy (mean±CI%95) over 600 few-shot tasks. All methods use parametric network family (pf) backbones.

|  | **SUR-pf** | **URT-pf** | **VS.** |
|---|---|---|---|
| ILSVRC | $56.0 \pm 1.1$ | $55.5 \pm 1.1$ | = |
| Omniglot | $90.0 \pm 0.8$ | $90.2 \pm 0.6$ | = |
| Aircraft | $79.7 \pm 0.8$ | $79.8 \pm 0.7$ | = |
| Birds | $75.9 \pm 0.9$ | $77.5 \pm 0.8$ | = |
| Textures | $72.5 \pm 0.7$ | $73.5 \pm 0.7$ | = |
| Quick Draw | $76.7 \pm 0.7$ | $75.8 \pm 0.7$ | = |
| Fungi | $49.8 \pm 1.1$ | $48.1 \pm 0.9$ | = |
| VGG Flower | $90.0 \pm 0.6$ | $\mathbf{91.9 \pm 0.5}$ | + |
| Traffic Signs | $52.2 \pm 0.8$ | $52.0 \pm 1.4$ | = |
| MSCOCO | $50.2 \pm 1.1$ | $52.1 \pm 1.0$ | = |
| MNIST | $93.2 \pm 0.4$ | $93.9 \pm 0.4$ | = |
| CIFAR10 | $66.4 \pm 0.8$ | $66.1 \pm 0.8$ | = |
| CIFAR100 | $57.1 \pm 1.0$ | $57.3 \pm 1.0$ | = |

## 4.4 URT USING FiLM MODULATED BACKBONES

As additional evidence of the benefit of URT on universal representations, we also present experiments based on a different set of backbone architectures. Following SUR (Dvornik et al., 2020), we consider the backbones from a parametric network family, obtained by training a base backbone on one dataset (ILSVRC) and then learning separate FiLM layers (Perez et al., 2018) for each other dataset, to modulate the backbone so it is adapted to the other domains. These backbones collectively have only 0.5% more parameters than a single backbone. More details of the backbones can be found in Appendix C.

A comparison between SUR and URT using these backbones (referred to as SUR-pf and URT-pf) is presented in Table 2. Once again, URT can improve the performance on VGG Flower without sacrificing performance on others.

## 4.5 HYPER-PARAMETER AND ABLATION STUDIES

We analyze the importance of the various components of URT's attention mechanism structure and training strategy in Table 3. First we analyze the importance of using the support set to model queries and/or keys. To this end, we consider setting the matrices $\mathbf{W}^q$ / $\mathbf{W}^k$ of the query / key linear transformation to 0, which only leaves the bias term. We found that the support set representation is most crucial for building the keys (row w/o $\mathbf{W}^k$ in the table) and has minor benefits for queries (row w/o $\mathbf{W}^q$ in the table). This observation is possibly related to the success of attention-based models with learnable constant queries (Liu et al., 2016; Lin et al., 2017). We also found that adding a regularizer $\Omega(\Theta)$ as in Equation 8 is important for some datasets, specifically VGG Flower and Birds.

Table 3: Meta-Dataset performance variation on ablations of elements of the URT layer.

|  | **ILSVRC** | **Omniglot** | **Aircraft** | **Birds** | **Textures** | **Draw** | **Fungi** | **Flower** | **Signs** | **MSCOCO** |
|---|---|---|---|---|---|---|---|---|---|---|
| w/o $\mathbf{W}^q$ | +0.2 | -0.2 | -0.6 | -0.1 | -0.3 | -0.2 | 0.0 | -0.2 | -0.8 | -0.1 |
| w/o $\mathbf{W}^k$ | -14.2 | -2.8 | -10.7 | -18.1 | -7.6 | -9.3 | -22.4 | -3.6 | -0.26 | -10.9 |
| w/o $r(S_c)$ | -14.2 | -2.8 | -10.7 | -18.1 | -7.6 | -9.2 | -22.4 | -3.6 | -0.26 | -10.9 |
| w/o $\Omega(\Theta)$ | 0.0 | -0.9 | -0.4 | -3.3 | -1.2 | -0.2 | +0.3 | -9.0 | -2.0 | 0.0 |

An important hyper-parameter in URT is the number of heads $H$. We chose this hyper-parameter based on the performance on validation set of tasks in Meta-Dataset. In Table 4, we show the validation performance of URT for varying number of heads. As suggested by Triantafillou et al. (2020), we considered looking at the rank of the performance achieved by each choice of $H$ for each validation domains, and taking the average across domains as a validation metric. However,

since the performances when using two to four heads are similar and yield the same average rank, we instead simply consider the average accuracy as the selection criteria.

Table 4: Validation performance on Meta-Dataset using different number of heads

|  | 1 | 2 | 3 | 4 | 5 | 6 | 7 | 8 |
|---|---|---|---|---|---|---|---|---|
| Average Accuracy | 74.605 | 77.145 | 76.943 | 76.984 | 76.602 | 75.906 | 75.454 | 74.473 |
| Average Rank | 2.875 | 1.000 | 1.000 | 1.000 | 2.250 | 2.250 | 2.25 | 2.50 |

In general, we observe a large jump in performance when using multiple heads instead of just one. However, since the number of heads controls the capacity, predictably we also observe that having too many heads leads to overfitting.

## 5 CONCLUSION

We proposed the URT layer to effectively integrate representations from multiple domains and demonstrated improved performance in multi-domain few-shot classification. Notably, our URT approach was able to set a new state-of-the-art on Meta-Dataset, and never performs worse than its predecessor (SUR) while also being $10\times$ more efficient at inference. This work suggests that combining meta-learning with pre-trained universal representations is a promising direction for new few-shot learning methods. Specifically, we hope that future work can investigate the design of richer forms of universal representations that go beyond simply pre-training a single backbone for each domain, and developing meta-learners adapted to those settings.

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

## A  EXPERIMENTS ON MORE DATASETS

We also report performances on the MNIST, CIFAR-10 and CIFAR-100 dataset sources in Table 5, and compare with the subset of methods that have reported on these datasets. There, URT neither improves nor gets worse performance than SUR, yeilding top performance on the MNIST domain but not on the CIFAR-10/CIFAR-100 domain, on which Simple CNAPS has the best performance.

Table 5: Test performance (mean+CI%95) over 600 few-shot tasks on additional datasets.

|  | MNIST | CIFAR10 | CIFAR100 | avg. rank |
|---|---|---|---|---|
| **CNAPs** | $92.7 \pm 0.4$ | $61.5 \pm 0.7$ | $50.1 \pm 1.0$ | 4.7 |
| **TaskNorm** | $92.3 \pm 0.4$ | $69.3 \pm 0.8$ | $54.6 \pm 1.1$ | 3.3 |
| **SUR** | $\mathbf{94.3 \pm 0.4}$ | $66.8 \pm 0.9$ | $56.6 \pm 1.0$ | 2.3 |
| **SimpleCNAPS** | $93.9 \pm 0.4$ | $\mathbf{74.3 \pm 0.7}$ | $\mathbf{60.5 \pm 1.0}$ | 1.7 |
| **URT (Ours)** | $\mathbf{94.8 \pm 0.4}$ | $67.3 \pm 0.8$ | $56.9 \pm 1.0$ | 2.0 |

## B  MORE RELATED WORKS

**Transfer by fine-tuning** A simple and effective method for few-shot classification is to perform transfer learning by first learning a neural network classifier on all data available for training and using its representation to initialize and then fine-tune neural networks on the few-shot classification tasks found at test time (Chen et al., 2019; Triantafillou et al., 2020; Dhillon et al., 2020; Saikia et al., 2020). Specifically, Saikia et al. (2020) have shown that competitive performance can be reached using a strong hyper-parameter optimization method applied on a carefully designed validation metric appropriate for few-shot learning.

## C  STRUCTURES AND TRAINING STRATEGY OF BACKBONES

Our URT layer can be built on top of a set of pretrained backbones. The structures of the backbones follow the approach in Dvornik et al. (2020). For Table 1, we use the ResNet18 architecture (He et al., 2016) for the backbones, where a separate backbone is pretrained on each domain separately using the corresponding training data in meta-dataset. The training domains include ImageNet, Omniglot, Aircraft, CU-Birds, Textures, Quick Draw, Fungi and VGG-Flower. For the parametric network family in Table 2, a *base* ResNet18 is first trained on ImageNet. Then a small number of modulating parameters are trained on the other domains, using their domain-specific training data, while the rest of the base backbone's weights stay fixed. Specifically, FiLM feature modulation (Perez et al., 2018) is used. This type of parametric network family thus allows for a much reduced number of learnable parameters, by reusing the weights from the base network. In experiments, we use the pretrained backbones released by Dvornik et al. (2020) for both cases, without

any further finetuning. End-to-end training of a set of backbones and the URT layers requires unaffordable computational cost, so we fix the pretrained backbones and only train the URT layer.

**Implementation details for training backbones** The training details of the backbones come from Dvornik et al. (2020). For optimization, SGD with momentum was used, using cosine learning rate annealing. Since the datasets come from different domains, the starting learning rate, the maximum number of training iterations and annealing frequency are set individually for each dataset. Data augmentation is applied and a constant weight decay of $7 \times 10^{-4}$ is set. For each dataset, a grid search over batch size in [8, 16, 32, 64] was run and the one that maximizes accuracy on the validation set was picked. For the parametric network family, the base ResNet18 trained on ImageNet is the same. For other backbones, cosine annealing as learning rate policy is also used, weight decay and data augmentation employed as above. Please refer to Table 4 and Table 5 in the original SUR paper (Dvornik et al., 2020) for the specific values of the hyperparameters for the individual feature networks and the parametric network family, respectively.

## D    RESULTS ON TRAFFIC SIGNS

The shuffle buffer bug described in meta-dataset issue #54 (https://github.com/google-research/meta-dataset/issues/54) has been propagated to previous works such as CNAPs (Requeima et al., 2019). The results for the Traffic Signs dataset are considerably worse after fixing this bug. For instance, URT degrades from 69.4±0.8 to 51.1±1.1. Please visit the official meta-dataset GitHub repo for more details. The main paper shows the corrected results (for URT and competing approaches). For completeness, we also provide the tables as they were in the initial version of this paper for your reference in this Appendix. Both sets of results overall support the advantageous performance of URT over previous work.

Table 6: Test accuracy (mean±CI%95) over 600 few-shot tasks. URT and the most recent methods, which are listed in the first column, are compared on Meta-Dataset (Triantafillou et al., 2020), which are listed in the first row. The numbers in **bold** have intersecting confidence intervals with the most accurate method.

| | ILSVRC | Omniglot | Aircraft | Birds | Textures | QuickDraw | Fungi | VGGFlower | TrafficSigns | MSCOCO | avg. rank |
|---|---|---|---|---|---|---|---|---|---|---|---|
| **MAML** | 37.8±1.0 | 83.9±1.0 | 76.4±0.7 | 62.4±1.1 | 64.1±0.8 | 59.7±1.1 | 33.5±1.1 | 79.9±0.8 | 42.9±1.3 | 29.4±1.1 | 8.0 |
| **ProtoNet** | 44.5±1.1 | 79.6±1.1 | 71.1±0.9 | 67.0±1.0 | 65.2±0.8 | 64.9±0.9 | 40.3±1.1 | 86.9±0.7 | 46.5±1.0 | 39.9±1.1 | 7.3 |
| **ProtoMAML** | 46.5±1.1 | 82.7±1.0 | 75.2±0.8 | 69.9±1.0 | 68.3±0.8 | 66.8±0.9 | 42.0±1.2 | 88.7±0.7 | 52.4±1.1 | 41.7±1.1 | 5.4 |
| **CNAPs** | 52.3±1.0 | 88.4±0.7 | 80.5±0.6 | 72.2±0.9 | 58.3±0.7 | 72.5±0.8 | 47.4±1.0 | 86.0±0.5 | 60.2±0.9 | 42.6±1.1 | 5.1 |
| **BOHB-E** | 55.4±1.1 | 77.5±1.1 | 60.9±0.9 | 73.6±0.8 | **72.8±0.7** | 61.2±0.9 | 44.5±1.1 | **90.6±0.6** | 57.5±1.0 | **51.9±1.0** | 4.4 |
| **TaskNorm** | 50.6±1.1 | 90.7±0.6 | 83.8±0.6 | 74.6±0.8 | 62.1±0.7 | 74.8±0.7 | 48.7±1.0 | 89.6±0.6 | 67.0±0.7 | 43.4±1.0 | 3.8 |
| **SUR** | 56.3±1.1 | 93.1±0.5 | **85.4±0.7** | 71.4±1.0 | **71.5±0.8** | 81.3±0.6 | **63.1±1.0** | 82.8±0.7 | 70.4±0.8 | **52.4±1.1** | 2.5 |
| **SimpleCNAPS** | **58.6±1.1** | 91.7±0.6 | 82.4±0.7 | **74.9±0.8** | 67.8±0.8 | 77.7±0.7 | 46.9±1.0 | **90.7±0.5** | **73.5±0.7** | 46.2±1.1 | 2.4 |
| **URT (Ours)** | 55.7±1.0 | **94.4±0.4** | **85.8±0.6** | **76.3±0.8** | 71.8±0.7 | 82.5±0.6 | **63.5±1.0** | 88.2±0.6 | 69.4±0.8 | **52.2±1.1** | 1.6 |

Table 7: Test accuracy (mean±CI%95) over 600 few-shot tasks. All methods use parametric network family (pf) backbones.

| | SUR-pf | URT-pf | VS. |
|---|---|---|---|
| ILSVRC | 56.4 ± 1.2 | 55.5 ± 1.1 | = |
| Omniglot | 88.5 ± 0.8 | **90.2 ± 0.6** | + |
| Aircraft | 79.5 ± 0.8 | 79.8 ± 0.7 | = |
| Birds | 76.4 ± 0.9 | 77.5 ± 0.8 | = |
| Textures | 73.1 ± 0.7 | 73.5 ± 0.7 | = |
| Quick Draw | 75.7 ± 0.7 | 75.8 ± 0.7 | = |
| Fungi | 48.2 ± 0.9 | 48.1 ± 0.9 | = |
| VGG Flower | 90.6 ± 0.5 | **91.9 ± 0.5** | + |
| Traffic Signs | 65.1 ± 0.8 | **67.5 ± 0.8** | + |
| MSCOCO | 52.1 ± 1.0 | 52.1 ± 1.0 | = |
| MNIST | 93.2 ± 0.4 | 93.9 ± 0.4 | = |
| CIFAR10 | 66.4 ± 0.8 | 66.1 ± 0.8 | = |
| CIFAR100 | 57.1 ± 1.0 | 57.3 ± 1.0 | = |

