# OpenReview forum: "A Universal Representation Transformer Layer for Few-Shot Image Classification"
_ICLR.cc/2021/Conference — ICLR 2021 Poster_

### Official Review · AnonReviewer2 · 2020-10-27
**This work proposed to meta-learn a module for self-attention to select between different domain-specific representation.**

**Rating:** 6
**Confidence:** 4

**Review:**

Summary: The paper proposes a meta-trained Universal Representation Transformer (URT) layer, which learn to dynamically re-weight domain-specific representation for classifying given target images. The evaluation on Meta-Dataset shows proposed method achieved competitive performance against compared baselines.

Strengths:
+ The proposed URT, inspired by self-attention and Transformer network, learned to dynamically  re0weight domain-specific representation for classifying images on an unseen target domain.
+ The proposed URT  can be used as a single-head URT layer  or a multi-head URT layer, where a regularizer (i.e., eqn (8)) is added to avoid duplicate attention scores in different URT layer.

Weakness:
- The idea of mixing pre-trained representation to a universal representation was first proposed in SUR. Compare to SUR, this work meta-trained a attention module for the mixing process (as compared to handcraft approach in SUR). While  this work shows good improvement over SUR with a learnable URT, I argue this work does not have sufficient theoretical or algorithmic contributions.
- The evaluation is only conducted on Meta-Dataset. Has the method evaluated on other widely-used domain generalization benchmarks, such as PACS and Office-Home.

Minor comments:
- Line 1 of the abstract "Few-shot classification aims to recognize unseen classes when presented with only a small number of samples" is incorrect. The statement is missing key information and misleading. There is an established literature on few-shot classification that learn a classifier with less samples (e.g, 5 sample per class) and the target is the learned classes. The author is probably refer to the few-shot domain generalization where the target class are from an unseen domain. But "unseen classes" is a fairly big statement to make in this case.
- Figure 1 shows the attention scores generate by URT.  For test domain ILSVRC, it assign high scores to Birds, followed by Fungi then ILSVRC.  Then, for test domain Birds and Textures, both assign high score to ILSVRC. I am curious why textures domain required higher attention to ILSVRC, but ILSVRC does not assign any score (or too low) to Textures. Birds' score seems to be more consistent. In other words, while URT attention heads generate interpretable score, how can we reason on the score and generate more insights on why a particular source domain is selected.

------------------
Post Rebuttal:
The authors' response has addressed my concerns. Based on the response and other reviewers' comment, I have updated my rating for this work.

---

> ### Author Response · Authors · 2020-11-13
> **Clarification on differences between few-shot classification and domain generalization**
>
> ### Responses to weaknesses:
> Q1: The idea of mixing pre-trained representation to a universal representation was first proposed in SUR. Compared to SUR, this work meta-trained an attention module for the mixing process (as compared to the handcraft approach in SUR). While this work shows good improvement over SUR with a learnable URT, I argue this work does not have sufficient theoretical or algorithmic contributions.
>
> A1: In our view, learning how to properly combine representations from multiple backbones is an important contribution and meaningful improvement over SUR. While this extension over SUR is natural, we feel that the successful implementation and rigorous demonstration of the merits of this approach is an important contribution to the field.
>
>
> Q2: The evaluation is only conducted on Meta-Dataset. Has the method evaluated on other widely-used domain generalization benchmarks, such as PACS and Office-Home.
>
> A2: The few-shot classification literature that this work belongs to and has followed builds on a tradition of evaluating tasks that correspond to new classes, not part of the training set of tasks. For example, see papers using smaller scale few-shot learning benchmarks Omniglot and *mini*ImageNet, wherein both cases the classes are separated into training classes and test classes [b,c,d]. Meta-Dataset is the largest scale, multi-domain benchmark falling under this framework. It is also the dataset used by SUR for its evaluation. Our understanding of PACS and Office-Home is that, while they are great benchmarks for domain generalization/adaptation, they however assume that the semantic identity of classes is the same at training and test time (only the input distribution under each class is changing).  If the reviewer thinks this clarification would be useful to be mentioned in the paper, we'd be happy to add it.
>
>
> ### Responses to minor comments:
>
> Q1: Line 1 of the abstract "Few-shot classification aims to recognize unseen classes when presented with only a small number of samples" is incorrect. The statement is missing key information and misleading. There is an established literature on few-shot classification that learns a classifier with fewer samples (e.g, 5 samples per class) and the target is the learned classes. The author probably refers to the few-shot domain generalization where the target class is from an unseen domain. But "unseen classes" is a fairly big statement to make in this case.
>
> A1: Please refer to our response 2 to the weaknesses above for the difference between a few-shot classification and domain generalization.
>
> Q2: Figure 2 shows the attention scores generated by URT. For the test domain ILSVRC, it assigns high scores to Birds, followed by Fungi then ILSVRC. Then, for test domain Birds and Textures, both assign high scores to ILSVRC. I am curious why the textures domain required higher attention to ILSVRC, but ILSVRC does not assign any score (or too low) to Textures. Birds' scores seem to be more consistent. In other words, while URT attention heads generate interpretable scores, how can we reason on the score and generate more insights on why a particular source domain is selected.
>
> A2: Figure 2 shows two head attention scores learned by URT. We found that for datasets from the seen domains, i.e. the first eight rows,  one head  (right,  orange)  consistently puts most of its weight on the backbone pre-trained on the same domain, while the other head (left, blue) learns relatively smoother weight distributions that blend other related domains. The reason why ILSVRC is often assigned with high weight probably is: this is a large dataset and the backbone trained on this dataset has better representation ability in general. More detailed explanations can be found in Section 4.3.
>
> [a] Triantafillou, Eleni, et al. "Meta-dataset: A dataset of datasets for learning to learn from few examples." ICLR 2020.
>
> [b] Finn, Chelsea, Pieter Abbeel, and Sergey Levine. "Model-agnostic meta-learning for fast adaptation of deep networks." ICML 2017.
>
> [c] Snell, Jake, Kevin Swersky, and Richard Zemel. "Prototypical networks for few-shot learning." Advances in neural information processing systems. 2017.
>
> [d] Vinyals, Oriol, et al. "Matching networks for one shot learning." Advances in neural information processing systems. 2016.

---

### Official Review · AnonReviewer4 · 2020-10-28
**No glaring issues**

**Rating:** 8
**Confidence:** 5

**Review:**

Summary

The paper presents a method for tackling multi-domain few-shot image classification problem where it obtains a task-adapted representation by weighing representations from pretrained domain-specific backbones according to the support set at hand. The desirable property of this framework is that the model can leverage information from other domains to make predictions. The effectiveness of Universal Representations have been discussed in the past work - SUR [1], and this work builds on top of it and introduces a learnable component (self-attention), and showed the improvement both quantitatively and qualitatively.


Strengths
- The paper is well-written
- The hypothesis is clearly conveyed, tested and is interpretable as seen from the attention weights
- The model improves over the results of the past works that were based on conditioning backbones using FiLM layers - CNAPs [2], Simple CNAPS [3]. While these past works have used additional modules such as a small CNN set encoder to encode task-representation, FiLM layers for conditioning; the simplicity and effectiveness of this model is appealing



Weaknesses

I have a high-level comment.

- Domain mixing during training:
    - If I recall correctly, the way sampling works in Meta-dataset is that a dataset domain is picked and then a task is sampled. Is there a way to try mixing domains in a task? I guess then the class-specific attention scores would vary a lot among classes (because some classes would prefer a different backbone that the other classes). So task-adapted representations would change to class-specific representations. And the only change in eq 9 and the expression of $p_c$ would be to replace $\phi(x)$ for a query image $x$ to $\phi_c(x)$
    - I don’t know if the above makes sense, but this will allow you to generalize to any real-world setting, and will also allow similar classes in different datasets to share information. Right now, the model is good at figuring out what domain does the task come from and find an appropriate mix of backbone for that task, however, what if it’s geared to do that for classes?


Minor concerns (suggestions, typos, etc.)
- Section 3.1
    - Mention dimensionality of weights and representations


Preliminary Rating and its justification
- I don’t see any glaring faults in this paper so I recommend accept.


[1] Nikita Dvornik, Cordelia Schmid, and Julien Mairal. Selecting relevant features from a universal representation for few-shot classification. arXiv preprint arXiv:2003.09338, 2020

[2] James Requeima, Jonathan Gordon, John Bronskill, Sebastian Nowozin, and Richard E Turner. Fast and flexible multi-task classification using conditional neural adaptive processes. In The Conference on Neural Information Processing Systems (NeurIPS), pp. 7957–7968, 2019

[3] Peyman Bateni, Raghav Goyal, Vaden Masrani, Frank Wood, and Leonid Sigal. Improved few-shot visual classification. In Proceedings of the IEEE Conference on Computer Vision and Pattern Recognition (CVPR), 2020.

---

> ### Author Response · Authors · 2020-11-13
> **Domain mixing during training**
>
> Thank you for reviewing our submission and the comprehensive comments!
>
> Q1: Domain mixing during training
>
> A1: Thank you very much for proposing the scenario where the samples in one task may come from different domains. This is a more practical setting and your proposal makes a lot of sense! We will look more into this direction in our future works!

---

### Official Review · AnonReviewer3 · 2020-10-30
**This paper proposes a transformer based exploitation of multiple domain-specific backbones to achieve better performance across all the domains at hand.**

**Rating:** 7
**Confidence:** 5

**Review:**

The review is brief because of time pressure. However, I have gone through the paper carefully.
Motivation
The paper is well motivated. It is keenly aware of previous work in the field and establishes its advancement of the state of the art clearly. It reviews past work in meta-learning as well as universal representations and transformers. I do have a suggestion for improvement, which is to consider the lifelong learning literature where reinforcement learning based methods have been developed for learning tasks over a lifetime. While reinforcement learning is a qualitatively different approach, lifelong learning requires the kind of adaptation to changes in tasks that the authors are addressing in their paper. It might behoove them to look at that literature and make a critical assessment with respect to their work. I don't see this as a weakness of the paper at all.

Approach
The approach is clearly described and is technically sound. It essentially sets up an optimization across multiple domain specific backbones to solve the multi-task problem. Such an approach has the advantage of modular design although I am curious to know if the authors have any opinions on how to introduce a new backbone into their system without having to retrain the entire system end to end. Or just in general how they would introduce a new domain specific backbone.
The optimization is clearly described and convincing.
Results
The results are convincing. They are at par or better than the state of the art. They are carried out on datasets well accepted by the community.
Quality, Clarity, Originality and Significance
Clarity - The paper is extremely well written. There are typos for example Representation is misspelled (misspelt). Those can be easily removed with a single editing pass. The paper motivates its approach well and describes the approach systematically. The results are presented convincingly and clearly. I would say the clarity of the paper is high.
Quality, Originality and Significance - The idea presented here is certainly novel in its details. The overall idea of using multiple domains to compensate for data-scarcity in certain domains is not new, but realizing that in a mostly better than the state of the art manner is a challenge that the authors address successfully. The overall proposal is a small but good idea that leads to good results. I would therefore say that the paper has good quality, significance and originality.

---

> ### Author Response · Authors · 2020-11-13
> **Extension to life-long learning and typos**
>
> Thank you for reviewing our submission under the pressure of a small-time window! We summarize your comments and response as below:
>
> Q1: Such an approach has the advantage of modular design although I am curious to know if the authors have any opinions on how to introduce a new backbone into their system without having to retrain the entire system end to end. Or just in general how they would introduce a new domain-specific backbone.
>
> A1: Thanks for this suggestion! We agree that this is an interesting direction for our future work! Some potential directions are a new design of query so that the dimension of $W^q$ does not condition on the number of backbones. For example, sampling a set of backbones instead of using representations for all backbones in every task. We also want to note that the training cost of our URT layer is small (only about 2 hours for training all datasets in meta-dataset). Thus, we believe retraining URT is affordable in practice.
>
> Q2: There are typos for example Representation is misspelled (misspelt)
>
> A2: We have corrected the typo and thanks for pointing it out!

---

### Official Review · AnonReviewer5 · 2020-11-03
**Review for A Universal Representation Transformer Layer for Few-Shot Image Classification**

**Rating:** 6
**Confidence:** 5

**Review:**

Summary
========
Few-shot learning on meta-dataset is challenging due to the domain gap between train and validation. In order to bridge this gap, the authors present a model that learns to combine domain-specific representations to generalize to new domains. This combination is done with a transformer model that pays attention to the features extracted from domain-specific backbones. The authors demonstrate empirically that their model attains comparable performance to previous state-of-the-art at higher efficiency and include ablation results to test their model components.

Overall Review
=============
The proposed method is sound and relevant for the research community. Although it lies towards the application side (i.e. a transformer on top of pre-trained features) and it has some weak points (see weaknesses), I still think that its simplicity will make it impactful. Thus, once the weaknesses are addressed I will happily raise my score.

Strengths
========
* The proposed method is simple and works well.
* The authors provide code and ablation experiments.
* The text is well-written and easy to follow.

Weaknesses
==========
* In understand that the model is more efficient because it does not need gradient descent at test time. Is that the case? if it is, could you include this information in the paper for completeness?
* Given that the proposed model is an efficient version of SUR, there are some questions that naturally come to the reader that are not answered in the current submission. For instance, how do the attention coefficients of URT compare with the coefficients learned by URT? Why does URT perform better on the held-out data? What is the difference in training time?

-------------------------------------
After Rebuttal
============
My main concerns were about the similarity between SUR and URT and the lack of detail in their comparison. I also asked for a clarification on the efficiency of the method.

On the first concern, they partially address it with the Coefficient characteristics, I say partially because I would have liked a more in-depth comparison of the characteristics, but technically they have addressed my question. For the second one, they now provide the training time, and the testing time could be found in Section 4.2.

Overall, even though I still think that this work lies in the application side, it is interesting enough to be published at ICLR, so I have accordingly raised my score.

---

> ### Author Response · Authors · 2020-11-13
> **Explanation of efficiency**
>
> Thank you very much for taking the time to review our submission and the comments! We are happy to respond to your comments as below:
>
> Q1: In understand that the model is more efficient because it does not need gradient descent at test time. Is that the case? if it is, could you include this information in the paper for completeness?
>
> A1: Yes, it is. Please refer to the last paragraph in Section 4.2 for related information: *Of note, the average inference time for URT is 0.04 second per task, compared to 0.43 for SUR, on a single V100.  Thus, getting rid of the optimization procedure for every episode with our meta-trained URT layer also significantly increases the latency, by more than 10×.*
>
> Q2: Given that the proposed model is an efficient version of SUR, there are some questions that naturally come to the reader that are not answered in the current submission. For instance, how do the attention coefficients of URT compare with the coefficients learned by SUR? Why does URT perform better on the held-out data? What is the difference in training time?
>
> A2:
>
> **Coefficients of URT and SUR:**
>
> - *Coefficient generation*: URT uses a fundamentally different way to aggregate information from the backbones. In particular, whereas SUR uses a fixed optimization procedure to separately weigh each backbone, URT learns how to mix and combine the information from the backbones using meta-learning.
> - *Coefficient formats*: The coefficients from SUR are one number for each backbone while URT can flexibly learn one coefficient per head for each backbone.
> - *Coefficient characteristics*: As we apply a regulariser on the attention scores as introduced in Section 3.2, every set of attention score for URT is regularised to be sparse and focus on a single domain while the scores from SUR do not have these characteristics.
>
> **Training time of URT and SUR:**
> Our URT layer can be built on top of a set of pretrained backbones. By only training the URT layer, the meta-training procedure only costs 2 hours on one V100 GPU. As mentioned in Section 4.2, the average inference time for URT is 0.04 second per task, compared to 0.43 for SUR, on a single V100.  Thus, getting rid of the optimization procedure for every episode with our meta-trained URT layer also significantly increases the latency, by more than 10×. We have updated Section 4.1 to include these discussions.

---

### Official Review · AnonReviewer1 · 2020-11-04

**Rating:** 7
**Confidence:** 5

**Review:**

## Summary

The paper addresses the problem of multi-domain few-shot image classification (where unseen classes and examples come from diverse data sources), and proposes a Universal Representation Transformer (URT) layer, which learns to transform a universal representation into task-adapted representations. The method proposed builds on top of SUR [Dvornik et al 2020], where a universal representation is extracted from the outputs of a collection of pre-trained and domain-specific backbones and a selection procedure infers how to weight each backbone for a given task at hand. While SUR inferred those weights by optimising a loss on the support set (the few examples provided in a task), the authors in this paper introduce an attention-based layer (inspired by Vaswani et al Transformer) that learns to weight the appropriate backbones for each task. This layer has the main advantage that it can be learned across few-shot tasks from many domains so it can support transfer across these tasks.


## Strengths

- The method and contributions are very well motivated and introduced. The paper is also very well written and very well presented. I also think that this new proposed URT layer is a very interesting contribution, and acknowledged its novelty for this specific task.

- The experimental section is good, which includes comparison with other state-of-the-art methods and an ablation study that analyses the contribution of the different components of the proposed approach. I find especially interesting section 4.3, where the attention scores produced by the network are visualised on the test tasks, which gives a better understanding of how this URT layer works.


## Weaknesses

- Architecturally, URT and SUR are pretty much identical, the only difference and novelty being the way the weights for the different backbones  are computed. This might affect the paper’s novelty impact.

- It would have been interesting to see how does SUR compare to URT with a single head, specially since the performance gap is quite significative from 1 to 2 layers as shown in Table 4. First, because it would give a deeper insight about the contribution of the different components of URT (attention layer vs multi-head). Second, because to me that’d be a bit more fair comparison between SUR and URT given that SUR only uses a single representation head: two heads means double dimensionality of the representation (from Eq 7, where representations are concatenated), and multi-head could also be applied to SUR using a similar approach (Eq 8).


Minor comments:

- It is not clear to me the claim done by the authors that SUR follows “hand-crafted feature-selection procedure” and that this procedure “is fixed and not learned”. If I understood correctly, SUR infers those weights by optimising a loss on the support set. While URT has a clear advantage since it doesn’t need to optimise on the support set, at the end of the day both infer those weights from the data, so how is it that SUR is hand-crafted? Apologies if I’m missing something, but I would like the authors if they could elaborate on this.


## Recommendation

Even though the method proposed doesn't differ too much from SUR, since the main difference is the way these weights are inferred, I still think that the new URT layer is an interesting contribution, and that paper brings enough novelty. For this reason, I’m initially leaning towards accepting the paper. However, I would like the authors to address my last two comments in the weakness section.

## After rebuttal

The authors have addressed my main concerns and I've decided to raise my score from 6 to 7.

---

> ### Author Response · Authors · 2020-11-13
> **Architectural comparison and explanation on multi-head**
>
> Thank you very much for taking the time to review our submission and the comments! Here are our responses to your comments:
>
> Q1: Architecturally, URT and SUR are pretty much identical, the only difference and novelty being the way the weights for the different backbones are computed. This might affect the paper’s novelty impact.
>
> A1:
> *Architectures for generating the weights:*
> In SUR, the weights for different backbones are learned by optimizing the loss on the support set for every task in the test stage. There is no shared model for these tasks and a set of learnable variables (with each for one backbone) are trained when evaluating each task. As for URT (ours), it meta-learns an attention-based layer on top of the backbones during the meta-training stage and can output the weight during the evaluation stage without any further training. A summary of the comparisons is:
>
> | | SUR | URT |
> | :---        |    :----  |          :--- |
> |Parameters for one task   |  a set of variables   &nbsp;&nbsp;     | dot product attention |
> |Parameters shared by all tasks?| No | Yes |
> |Meta-train?| No | Yes |
> |Need additional training in the evaluation stage?&nbsp;&nbsp;&nbsp;&nbsp;&nbsp;| Yes | No|
>
> *Architectures for backbones / feature extractors:*
> URT can be built on top of any pretrained backbones, we show the applicability and advantage on two sets of backbones as in Table 1 (ResNet18) and Table 2 (FiLM modulated ResNet18). We claim our contribution does not lie in the structure of the backbones but in how to generate the weights as above.
>
> Q2: It would have been interesting to see how SUR compares to URT with a single head, especially since the performance gap is quite significant from 1 to 2 layers as shown in Table 4. First, because it would give a deeper insight into the contribution of the different components of URT (attention layer vs multi-head). Second, because to me that’d be a bit more fair comparison between SUR and URT given that SUR only uses a single representation head: two heads mean double dimensionality of the representation (from Eq 7, where representations are concatenated), and multi-head could also be applied to SUR using a similar approach (Eq 8).
>
> A2: Thanks for this suggestion! In SUR, the representation is a concatenation of the features from different backbones. The representation dimension is $m\*f_d$, where $m$ is the number of backbones ($m=8$ for meta-dataset) and $f_d$ is the dimension of the feature generated by one backbone ($512$ for ResNet18). The representation dimension from URT as shown in Eq.7 is $H*f_d$, where $H$ denotes the number of heads. Thus, while we did find that having multiple heads was necessary to achieve strong performance, it's important to note that SUR already has "multiple heads" and more representation dimensions due to its use of concatenation. The weights inferred by SUR are not constrained to sum to one, unlike the weights produced by one attention head. This is why SUR is, by construction, multi-headed. Also, please refer to Figure 2 which visualises the separate contribution of the two attention heads.

---

### Decision · Program_Chairs · 2021-01-07
**Final Decision**

**Decision:**

Accept (Poster)

**Comment:**

This paper studies the problem of multi-domain few-shot image classification and proposes a Universal Representation Transformer (URT) layer, which leverages universal features by dynamically re-weighting and composing the most appropriate domain-specific representations in a meta-learning way. The paper extends the prior work of SUR [Dvornik et al 2020] by using meta-learning and avoiding additional training during test phase. The experimental results show improvements over SUR in both accuracy (not always significant on some datasets though) and inference efficiency. Overall, the paper is well written with sufficient contributions. After the author's rebuttal and revision, reviewers generally agree the paper can be accepted. I recommend to Accept (Poster).